# Snow depth derived from Sentinel-1 compared to in-situ observations in northern Finland

Adriano Lemos[1], Aku Riihelä[1]

[1] Finnish Meteorological Institute, Helsinki, Finland

*Correspondence to*: Adriano Lemos (adriano.lemos@fmi.fi)

**Abstract**

Seasonal snow in the northern regions plays an important role providing water resources for both consumption and hydropower generation. Moreover, the snow changes in northern Finland during winter impact the local agriculture, vegetation, tourism and recreational activities. In this study we estimated snow depth using an empirical methodology applied to the dual-polarisation of the Sentinel-1 synthetic aperture radar (SAR) images and compared with in situ measurements collected by automatic weather stations (AWS), and snow courses in northern Finland. We applied an adapted version of the empirical methodology developed by Lievens et al. (2019) to retrieve snow depth, using Sentinel-1 constellation between 2019 and 2022, and then compared to measurements from three automatic weather stations available over the same period. Overall, the Sentinel-1 snow depth retrievals were underestimated in comparison with the in-situ measurements from the automatic weather stations. We found slightly different patterns for the different years, and an overall correlation factor of 0.41, and a higher correlation in the 2020–2021 season (R=0.52). The high correlation between estimated and measured snow depth at the Inari Nellim location (R=0.81) reinforces the potential ability to derive snow changes in regions where in situ measurements of snow are currently lacking. Further investigation is still necessary to better understand how the physical properties of the snowpack influence the backscatter response over shallow snow regions.

## 1 Introduction

Snow variations play an important role in the northern regions, providing water resources for both consumption and hydropower generation. Seasonal snow variations in northern Finland during winter impact the local agriculture, vegetation, tourism and recreational activities (Lehtonen et al., 2013; Luomaranta et al., 2019). Some regions in the Arctic are experiencing a shortening in the snow cover duration during the past decades, and future projections demonstrate an increase in the surface temperature and a continuous decrease of snow cover through time for the northern regions of Finland (Lehtonen et al., 2013; Luomaranta et al., 2019). Thus, extensive monitoring of snow depth is crucial for various purposes.

Different measurements efforts play an important role in monitoring snow depth, including the Automatic Weather Stations (AWS; Luomaranta et al., 2019), light detection and ranging (LiDAR) flights (Painter et al., 2016), and snow course measurements (Leppänen et al., 2016). The collection of these data provides valuable and accurate measurements. However, their spatiotemporally limited coverage restricts systematic monitoring. On the other hand, remote sensing techniques, such as satellite observations and modelling, are key to improve the monitoring of snow over large areas all year around (Tsai et al., 2019; Awasthi & Varade, 2020; Tsang et al., 2022). Satellites equipped with passive microwave radiometry sensors, supported by the in situ measurements, have been extensively used to estimate snow water equivalent (SWE), the total water content in the snowpack, for decades (Takala et al., 2011; Pulliainen et al., 2020). However, despite their daily temporal resolution, the coarse spatial resolution (approximately 25 km by 25 km) and the dependency on the in-situ measurements still impose some limitations on the use of passive microwave radiometry for snow cover monitoring.

Currently, several studies in shallow snow regions, where snow thickness is lower than 1 m, make use of the synthetic aperture radar (SAR) measurements in the Ku-band (~ 12 – 18 GHz), as well as the Ka-band (~ 26.5 – 40 GHz), as these frequencies are more sensitive to snow pack changes. However, the exact knowledge of the penetration depth of the SAR signal in the snow pack still remains unknown and dependent on assumptions due to the snowpack characteristics, hindering accurate assessments (Tsang et al., 2022; Jutila and Hass, 2023).

The use of Interferometric Synthetic Aperture Radar (InSAR) technique using the L-band (~ 1 – 2 GHz) has shown promise, as it operates at lower frequencies and is less affected by the presence of vegetation and dry snow (Ruiz et al., 2022). However, the lack of freely available data makes its use more difficult. Future missions, such as the Radar Observing System for Europe in L-band (ROSE-L), as well as the NASA-ISRO Synthetic Aperture Radar (NISAR), will provide freely available L-band data worldwide, improving our understanding of snow changes and improving its monitoring capabilities.

The C-band backscatter measurements are widely used in several applications in the cryosphere. More specifically in the context of snow research, previous studies explore the application of the SAR images to provide information of dry snow

accumulation (Bernier and Fortin, 1998), and evaluation of snowmelt dynamics in the alpine regions (Marin et al., 2020). The behaviour of the C-band backscatter inside the snowpack is complex, and still an ongoing area of investigation (Hoppinen et al., 2024). Previous studies show that backscatter variations during mid-winter for shallow snow regions are dominated by the snow-ground interface and the dielectric constant of the soil, minimising the effect of the dry snowpack (Sun et al., 2015). However, minimal changes in the snow microstructure, and in the water liquid content in the snowpack, impacts the surface and volume scattering of the snow (Lievens et al., 2019, 2022). Despite some challenges and limitations, the use of the C-band (5 – 6 GHz) synthetic aperture radar images have demonstrated the ability to estimate snow depth and provide valuable information about snow depth variations using the Sentinel-1 (S1) constellation (Lievens et al., 2019, 2022; Dunmire et al., 2024; Hoppinen et al, 2024). They demonstrated the sensitivity of the co- and cross-polarised backscatter observations from the S1 satellites to estimate snow depth over mountainous regions in the Northern Hemisphere, where the snow thickness exceeds 1 m. These findings open the potential and significance of the use of the Sentinel-1 SAR images archive to estimate snow depth variation.

Snow depth estimates with high spatio-temporal resolution can improve our understanding of seasonal snow mass in complex access areas. Thus, the objective of this study is to expand the use of the empirical methodology applied to synthetic aperture radar images (Lievens et al., 2019) to estimate seasonal snow depth variations over shallow snow regions, in northern Finland. The findings will then be compared with independent in situ measurements collected by automatic weather stations (AWS), and snow courses, in the same area.

**2 Data and methods**

**Study Area**

The study area is located in the northern region of Finland, between the latitudes 68.3° and 69.3°N (Figure 1). The study area has a relatively flat topography, ranging approximately between 100 m to 500 m in elevation. The snow depth (SD) fluctuation is influenced by the variation of the local surface air temperature and precipitation (Luomaranta et al., 2019). In the northern part from 1961–2014 the average snow depth during winter was 82.7 cm, and maximum snow depth reached 121.5 cm in 2000 (Luomaranta et al., 2019). Due to its proximity, the temperature variations in Northern Finland have a strong influence of the Arctic Ocean (Aalto et al., 2016). The mean surface temperature in the north during the winter from 1988–2014 was -11.1°C, and average maximum surface temperatures reached approximately -7.2°C during the winter for the same period (Luomaranta et al., 2019).

**Automatic weather stations**

In order to compare and evaluate the snow depth estimates derived from S1, we used snow depth and surface air temperature measurements from three automatic weather stations (AWS), managed by the Finnish Meteorological Institute. The snow depths are measured by the Campbell Scientific SR50AH instruments mounted on the stations, and the instrument accuracy, according to the manufacturer, is approximately 1 cm. We extracted information of daily snow depth and surface air temperature, spanning from 2019 to 2022, from the Finnish stations database around the Inari Lake (IL) region. The available AWS's, followed by their respective locations (Figure 1), elevation in meters above sea level (m.a.s.l.), and percentage of forest cover (FC) extracted from the Multi-source National Forest Inventory Raster Maps of 2021 described below) are; Inari Nellim (IN - 68.849°N, 28.399°E, 121 m.a.s.l., 33% of FC), Inari Kaamanen (IK - 69.141°N, 27.266°E, 158 m.a.s.l., 26% of FC), and Inari Angeli Lintupuoliselkä (IA - 68.903°N, 25.736°E, 240 m.a.s.l., 24% of FC).

**Snow courses**

There are approximately 140 snow courses across Finland. Snow course measurements are operated, and provided, by the Finnish Environment Institute (SYKE). Systematic measurements have been made, for some locations, by SYKE and the Finnish Meteorological Institute (FMI) since the 1930s (Leppänen et al., 2016). Typically, each snow course is 2 to 4 kilometers long, measured in the middle of each month, and at about 80 regularly spaced points, usually every 50 meters along the route (Leppänen et al., 2016). In this paper, we used averaged snow depth measurements along 6 snow courses (Figure 1); Inari Nellim (IN - 68.849°N, 28.399°E), Inari Angeli Lintupuoliselkä (IA - 68.903°N, 25.736°E), Inari Mutusjärvi (IM - 68.961°N, 26.739°E), Inari Repojoki (IR - 68.450°N, 25.977°E), Inari Kaamasmukka (IKa - 69.307°N, 26.656°E), and Inari Laanioja (IL - 68.371°N, 27.453°E).

**Canopy cover**

We used the canopy cover from the Multi-source National Forest Inventory Raster Maps of 2021 (MS-NFI), which is processed and distributed by the Luonnonvarakeskus (Natural Resources Centre) from Finland, to evaluate the correlation with the snow depth patterns derived from S1. The main products used to derive the canopy cover, and the other products distributed, are from the Sentinel-2A/B satellites of European Space Agency (ESA) and the Landsat 8 satellite of United States Geological Survey (USGS), the full description of the data is found in Mäkisara et al. (2022). The dataset comes in the ETRS-TM35FIN coordinate system, and the spatial resolution is posted at 16 m by 16 m. Areas affected by cloud coverage, regions outside forest land, and outside Finland are removed and disregarded (Mäkisara et al., 2022).

**Sentinel-1 data**

In this study we estimated snow depth using single look complex (SLC) synthetic aperture radar images acquired in the interferometric wide swath (IW) mode from the S1a satellite launched by the European Space Agency (ESA) in October 2014. Sentinel-1b was launched in April 2016 and ended its mission in December 2021 due to technical issues. For this reason, in the present work, we preferred to use only images acquired from Sentinel-1a, and referred from here as S1. The Sentinel SAR instruments operate at C-band (5.405 GHz), and the IW mode has a 250 km swath and spatial resolution of 5 m in ground range and 20 m in azimuth. Each satellite from the S1 constellation had a repeat cycle of 12 days and 180 degrees orbital phasing difference. We used the dual-polarisation (VH and VV) components from 56 SAR S1 images acquired over the same region in northern Finland. The data range acquired spans from October 2019 to May 2022 (Table S1 in the Supplementary data), and we followed the workflow described below to derive 56 snow depth maps.

In the pre-processing stage we used ESA's Sentinel Applications Platform (SNAP) software (version 8.0). We performed a standard processing routine for all the S1 SLC IW images, including the application of the most recent orbit file, radiometric calibration, deburysting and range-Doppler terrain correction using the Copernicus digital elevation model (DEM) posted to a spatial resolution grid of 30 m. Previous studies showed that speckle noise makes the data product more variable, and the upscaling of the S1 data has presented better snow depth estimates (Lievens et al., 2022; Dunmire et al., 2024; Hoppinen et al., 2024). In order to reduce speckle noise in the SAR measurements, we applied a moving mean filter to the data, using a kernel of 990 m by 990 m. The final pre-processed product was a time-series of stacked S1 images with $\sigma^0$ backscatter intensities in decibel (dB) for both HV and VV.

We used an adapted version of the empirical methodology developed by Lievens et al. (2019) to estimate snow depth using S1 products (Equations 1 and 2). The algorithm utilises changes in the cross-polarized backscatter measurements of SAR images repeatedly acquired on the same location and orbit to avoid geometry distortions. We calculated the ratio between the two cross-polarised ($\sigma^0_{vh}$ and $\sigma^0_{vv}$) backscatter intensities (in dB) in a pixel scale for the entire image time-series. We considered the entire region as susceptible to snow accumulation, and the snow index (SI) in the time step $t_i$, was calculated as described in the Equation (1). Moreover, if $SI(t_i) < 0$, it was considered as zero.

$SI(t_i) = SI(t_{i-1}) + [(\sigma^0_{vh}/\sigma^0_{vv})(t_i) - (\sigma^0_{vh}/\sigma^0_{vv})(t_{i-1})]$            (Equation 1)

The translation to snow depth (SD), in metres, is then calculated using Equation 2.

$SD(t_i) = \left(\dfrac{a}{1 - bFC(i)}\right) SI(t_i)$            (Equation 2)

156

The parameter a=1.1 m dB$^{-1}$ (Equation 2) is constant and was estimated using in situ measurements, minimising the mean absolute error (MAE) between the times series of the global average snow depth measurements and S1 estimates in mountain regions (Lievens et al., 2019). The forest cover (FC) used here is the canopy cover from the Multi-source National Forest Inventory Raster Maps of 2021 (MS-NFI). As the canopy cover attenuates the backscatter from the snow, an additional parameter b=0.6 (dimensionless), estimated by Lievens et al. (2019), is applied.

162

Errors in our snow depth estimates arise mainly through the radiometric accuracy for S1, specified as ~1 dB (Torres et al., 2012). Due to the fact we averaged all the $\sigma^0$ images to reduce speckle, an additional 0.5 dB was considered into the overall radiometric accuracy (Torres et al., 2012). The resulting radiometric accuracy of 1.5 dB, representing ~10-15% of the $\sigma^0$ signal, was used to determine the uncertainty of the snow depth measurements.

167

168

**3 Results and Discussions**

170

We used the S1 dataset (Table S1) between 2019–2022 to produce up-to-date snow depth at our designated study area (Figure 1). To explore changes in snow depth over space and time, we further extracted time series of snow depth to compare them to independent measurements from the three automatic weather stations (Figure 2). Then, we show mean snow depths yearly in Figure 3. Figure 4 presents the snow depth estimates separated by canopy density intervals. Furthermore, in order to evaluate the snow depth estimates from S1, the dataset was compared to the automatic weather stations in different scenarios, presented in the Figures 5 and 6.

177

Figure 2 displays the seasonal changes in the snow depth over three consecutive winters at the AWS sites. We observe that the snow depth estimates from S1 at the Inari Nellim location (Figure 2a) follows the seasonal variations measured by the automatic weather stations measurements, despite the underestimated values. The snow depth products derived from S1 from the other weather stations, IK and IA (Figures 2b and 2c), also follow the seasonality of the weather stations measurements, although they exhibit an evident underestimation relative to the AWS measurements. Automatic weather stations are usually located in relatively flat and non-forested terrain, which may not accurately represent the surrounding area, susceptible to changes in e.g., forest cover and terrain. Thus, it is important to highlight the challenges when comparing observations from a point-scale measurement from the AWS's, and the grid-scale estimates from S1 (Lievens et al., 2022). For this purpose, we compared the snow depth estimates from S1 to average snow depth measured (Figure S3) along the snow courses at 6 locations (Figure 1) available for the region. Overall, we observed underestimations in the snow depth estimates (Figure 2 and S3). Theoretically, the underestimation is possibly due to the water content in the snowpack, reflecting and absorbing the backscatter signal, as the ground temperature in the accumulation period remains approximately the same, insulated by the

snow (Lievens et al., 2019; Marin et al., 2020). The mean snow depths from S1 estimates are ~20.0 cm, ~10.1 cm, and ~13.4 cm, for Inari Nellim, Inari Kaamanen, and Inari Angeli L. locations respectively (Table 1). In contrast, the mean snow depth measured by the automatic weather stations IN, IK and IA are, respectively, ~37.1 cm, ~46.9 cm, and ~44.9 cm (Table 1). We notice from Figure S1, presenting the bias evolution of the snow depth as a function of the days of the year, that the snow season onset is well estimated by the method, despite the rapid bias increase as the snow season progresses.

The maps in Figures 3 present the average snow depth along the years. Overall, we find higher mean snow depth estimates in 2019–2020 (Figure 3a), following the AWS's measurements from the time series in Figure 2 during the same year. Furthermore, we noticed higher mean snow thickness over water bodies regions, reaching values over 50 cm for all the estimates along the years (Figure 3). In order to compare the snow thickness estimates from S1, we plotted the snow depth measured in snow pits (sp1-4 in Figure 1) during a field campaign around the Inari Lake region from the 3rd to 7th of April 2022 against the estimates 6th of April 2022 from S1 (Figure S2), as this is the closest estimate to the field measurements. We observe that, in comparison with the snow pits measurements on the lake region, all the snow depth derived from S1 are overestimated (Figure S1). Moreover, visually comparing the backscatter signal from the co- and cross- polarizations, VV and VH respectively, from S1 (Figures S4 and S5), we can observe that the VV component demonstrates to be more sensitive when the lake starts freezing, around 11th November. The backscatter signal increases (Figures S4 and S5), leading to an increase in the snow depth values.

Forest areas attenuate the radar waves, scattering the emitted and the received signal from the satellite to the snow cover on the ground, and vice-versa, leading to an underestimation of the results (Lievens et. al, 2019; Tsang et al., 2022). In order to investigate the influence of the forest cover, we divided the canopy density map (Figure 4a), from Multi-source National Forest Inventory Raster Maps of 2021, into forest cover density intervals and calculated the mean snow depth for each interval yearly (Figure 4b). We observe for all the years, and overall mean, thicker snow depth values over dense vegetation (50-100% of canopy coverage) and water bodies areas (Figure 4b). The mean snow depth from the year 2021-2022 (red bars in Figure 4) presents a slight snow depth decrease where the canopy density is above 40. For the 2019-2020 and 2020-2021 years, we found thicker snow layers over denser canopy regions (orange and green bars in Figure 4b, respectively). Despite the aligned increase of snow thickness and canopy density, the estimated snow depth over the forested areas are underestimated if compared to the automatic weather stations (Figure 2). Figure 4b shows a maximum snow depth of ~57 cm (canopy density over 20%) in 2019-2020, and a maximum snow depth of ~37 cm for the remaining years. Similar results were found using L-band SAR images, showing that the snow depth variations over the forested areas are also underestimated compared to vegetation free regions (Ruiz et al., 2022). It is important to comment that we also utilised the same approach described before (Figure 4) to correlate our snow depth estimates with terrain elevation intervals. We divided the digital elevation model in intervals every 100 m, going up to its maximum (~500 m). However, we have not found any significant correlation to include in this manuscript.


In order to compare the S1 estimates and the AWS's measurements, we calculated the temporal correlation coefficients in two
different scenarios (Figs. 5 and 6). In the first scenario (Sc1) we considered all the measurements at once, as well as separated
AWS's locations (Figure 5). In the second scenario (Sc2), we looked at individual years separately (Figure 6). Figure 5 displays
the overall correlation, Sc1, using all the 174 measurements for all the years and from the three sites. It presented a low
correlation of 0.41 and a mean absolute error of ~26.1 cm (Table 2). The estimates at the Inari Nellim weather station had a
high correlation of 0.81, when compared with the other locations with R=0.09 and R=0.55 for Inari Kaamanen and Inari Angeli
locations, respectively (Figure 5). Figure 6 presents all the 174 measurements separated yearly. We observe that the year 2020–
2021 had the higher correlation factor, R = 0.52, as well as the smaller mean absolute error (~15 cm; Table 2). The years 2019–
2020 and 2021–2022 presented correlation factors of 0.29 for both years (Figure 6), and mean absolute errors of ~38.9 cm and
~25.5 cm, respectively (Table 2).

The uncertainty in the AWS snow depth observations (~1 cm) is considerably smaller than the uncertainty of the SAR-based
estimates due to radiometric noise in the SAR imagery. At the Nellim site, a considerable part of the bias between the SAR-
based estimate and ground truth could be explained by the estimation uncertainty, yet the same does not hold for either
Kaamanen or Angeli. We thus conclude that the observed underestimation should be considered significant in relation to the
uncertainty of the estimation method.

The backscatter signal from co-polarised images in the C-band on dry snow conditions is strongly influenced by the ground
underneath, and by the water content in the snowpack (Sun et al., 2015; Marin et al., 2020; Feng et al., 2021; Lievens et al.,
2022). ERS and Radarsat, both in the C-band, demonstrated an increase in the co-polarised backscatter signal during the snow
accumulation periods (Bernier and Fortin, 1998) and a decrease over shallow areas (Rott and Nagler, 1993). Following the
same empirical hypothesis demonstrated by Lievens et al. (2019) and Lievens et al. (2022), the cross-polarised backscatter
signals at C-band are more responsive to dry snow accumulation, in comparison to the backscatter influence from the ground.
Lievens et al. (2019) suggest that dry snow is represented by layers of large clusters of irregular ice crystals, scattering on the
snow layer interfaces. Therefore, for deep snow locations, it is expected that layered snow enhances and dominates the
backscatter signal, from cross-polarised observations (Lievens, et al., 2019).

Given the considerable underestimation of snow depth over land, and conversely considerable overestimation of snow depth
over lake ice, our results reinforce the idea that the EM properties of the surface underlying the shallow seasonal snowpack
likely play a major role in the observable SAR backscatter. There is a clear need for dedicated studies to improve radiative
transfer modelling of volume scattering of snow in order to better explain the observed behaviour, as pointed out by Lievens
et al. (2019). Finally, it is worth pointing out that the backscatter ratios are converted into snow depth through empirical
coefficients. While the calibration coefficients are based on a large number of data (Lievens et al., 2019), they are based on
relationships observed for mountainous snow packs, and thus not necessarily valid for shallow snow packs elsewhere.
Recalibration of the coefficients is not considered here due to the limited number of reference snow depth observation sites in
our study area. We also point out that at Kaamanen in particular, the temporal evolution of the backscatter ratios would not
have tracked the snow depth evolution even if other linear calibrations were attempted. This further points to a need for rigorous
radiative transfer studies to better understand the composition of C-band SAR backscatter over seasonal shallow snowpacks.
**4 Conclusions**
We investigated the use of co- and cross-polarised backscatter from Sentinel-1 SAR C-band images to estimate snow depth
variations over the northern region of Finland from 2019 to 2022. We presented a high temporal resolution comparison between
snow depth estimated from S1 images and measurements from automatic weather stations, and correlated with canopy cover
provided by Luonnonvarakeskus (Natural Resources Institute of Finland). The use of the C-band SAR to estimate snow depth
over shallow snow regions presented limitations. In general, we found underestimation for all the years and locations. It is
important to highlight the snow depth estimates at the Inari Nellim location, which demonstrated the best results (R=0.81),
when compared to the automatic weather station measurements at the same location. Looking throughout the years, the year
2020–2021 presented better results (R=0.52), when compared to the previous years.
We also investigated the correlation between the canopy coverage and the snow depth estimations, and we observed thicker
snow depth values over dense vegetation and water bodies regions. These findings are possibly due to the high sensitivity of
the VV component over freshly frozen water, increasing the backscatter significantly. We recognize that deriving shallow
snow depths using C-band SAR images is still a challenge and further investigation is necessary to better understand the
observed underestimation. Thanks to the effort of international space agencies, we have available currently, and will have in
the near future, global coverage at high-temporal and -spatial resolution of SAR imagery. Combined with installed automatic
weather stations, this opens the possibility of a wide spatial monitoring of snow variations independent of weather or solar
illumination conditions. However, given the present under- and overestimations observed against reference snow depth data,
we emphasise the first-order need for rigorous radiative transfer model-based studies to comprehensively understand the
drivers of SAR backscatter from snowpacks.
*Data availability*. The Sentinel-1 data are freely available at https:// https://browser.dataspace.copernicus.eu/ (last access: 09
July 2025). The automatic weather station datasets, provided by the Finnish Meteorological Institute, are available at
https://hav.fmi.fi/ (last access: 02 February 2023). The National Forest Inventory Raster Maps of 2021, provided by the
Luonnonvarakeskus (Natural Resources Institute of Finland), is available at https://kartta.luke.fi/ (last access: 09 July 2025).

The snow courses measurements, provided by Suomen ympäristökeskus (Finnish Environmental Institute), are available on request (details in http://litdb.fmi.fi/manual_measurements.php). The processed data is available on request to the corresponding authors and will be available on the METIS - Finnish Meteorological Institute Research Data repository, 10.57707/fmi-b2share.4de929068a064dfab78e5b9eeef79ce9.

*Author contribution:* AL and AR designed the study. AL led the processing and data analysis, and wrote the main manuscript. AR contributed to writing the manuscript and interpreting the results.

*Competing interests*. The authors declare that they have no conflict of interest.

*Acknowledgements.* This work was supported by the Academy of Finland, under the project "Low orbit altimetry, albedo, and Antarctic Snow and Sea-ice Surface Roughness" (LAS3R), grant number 335986. The authors gratefully acknowledge the European Space Agency for the Copernicus Sentinel 1 data acquired from Data Hub, courtesy of the EU/ESA. The Sentinel-1 data are freely available at https:// https://browser.dataspace.copernicus.eu/ (last access: 09 July 2025). The authors also acknowledge Luonnonvarakeskus (Natural Resources Institute of Finland) to provide the National Forest Inventory Raster Maps of 2021, and the Finnish Meteorological Institute for the automatic weather station datasets available at https://hav.fmi.fi (last access: 02 February 2023). We acknowledge Pjnia Venäläinen and Kari Luojus for providing the snow courses dataset. We thank the editor Ruth Mottram, and the two anonymous referees, for their comments, which helped to improve the manuscript.

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

370

371

**Figures**

Figure 1: Average snow depth estimated from S1 between 2019–2022 (between October and March). Black triangles indicate the automatic weather stations' locations; Inari Nellim (IN), Kaamanen (IK), and Angeli Lintupuoliselkä (IA), respectively. The red dots are representing the snow pits measurements (sp1–sp4). Yellow circles are the snow course locations; Inari Angeli Lintupuoliselkä (IA), Inari Kaamasmukka (IKa), Inari Laanioja (IL), Inari Mutusjärvi (IM), Inari Nellim (IN), and Inari Repojoki (IR). The inset figure shows the study region in Finland.

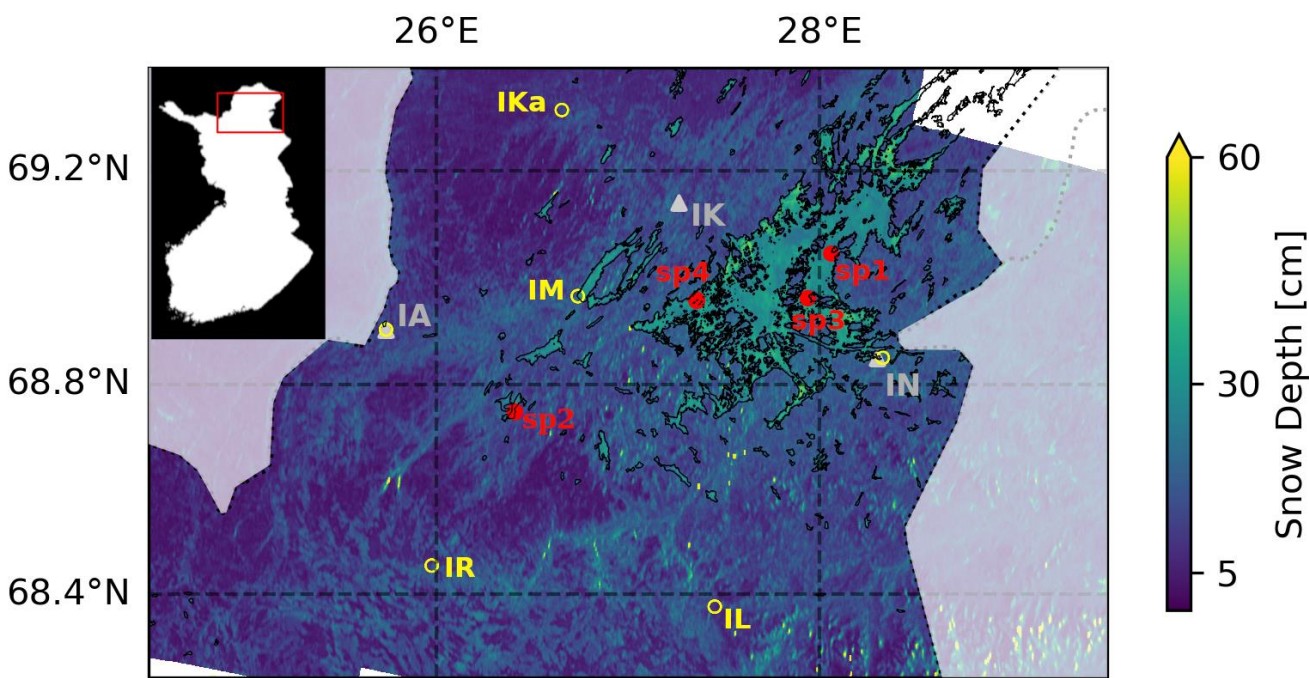

Figure 2: Snow depth variation between 2019 and 2022. The blue dots represent the snow depth variation estimated from the
S1 images before the correction done due the calibration and forest cover (FC) attenuation. Corrected values are represented
by the red dots. The uncertainty ranges are represented by the light blue and red shading. On the left y-axis, the solid black
line represents snow depth from the automatic weather stations and the blue dots are snow depth estimates derived by S1. On
the right y-axis, the solid red lines represent surface temperature daily averaged respectively.


Figure 3: Average snow depth estimated from S1 during the years of 2019–2020 (a), 2020–2021 (b), and 2021–2022 (c), respectively.

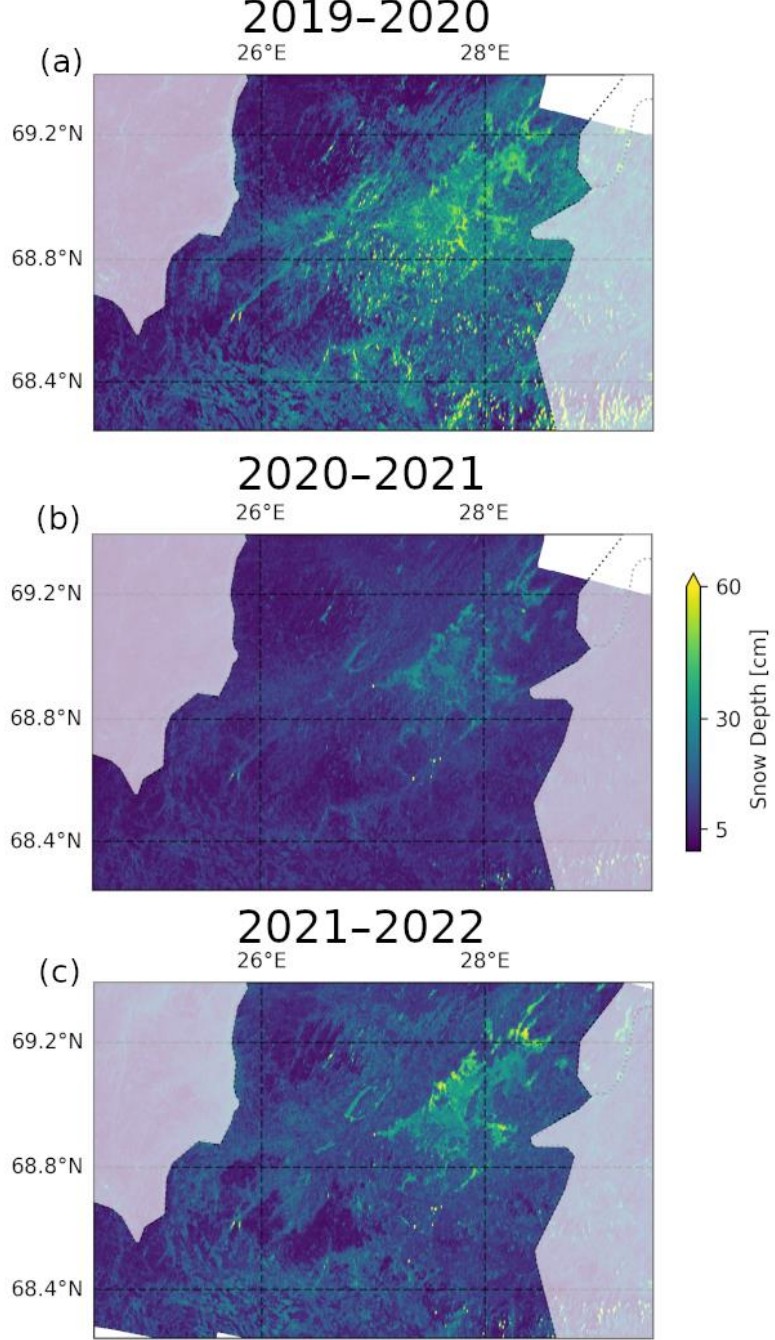

Figure 4: Canopy density map represented from 2021 (a). Mean snow depth separated in different canopy density intervals (b). The bottom and top of the vertical boxes represent the 25th and 75th interquartile, respectively. The solid black line inside the boxes represents the median snow depth estimate for each interval. Values outside the whiskers' extent are not shown and they are statistically considered outliers.

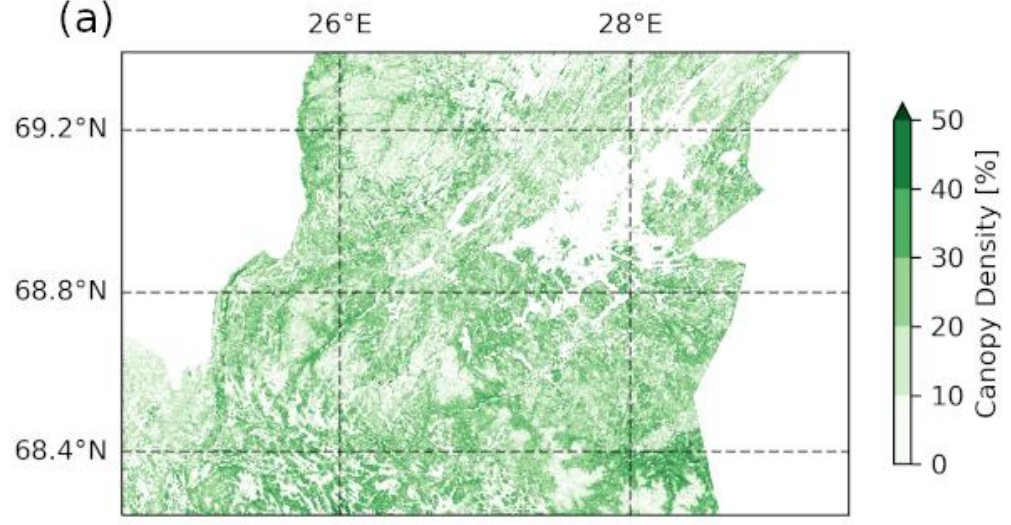

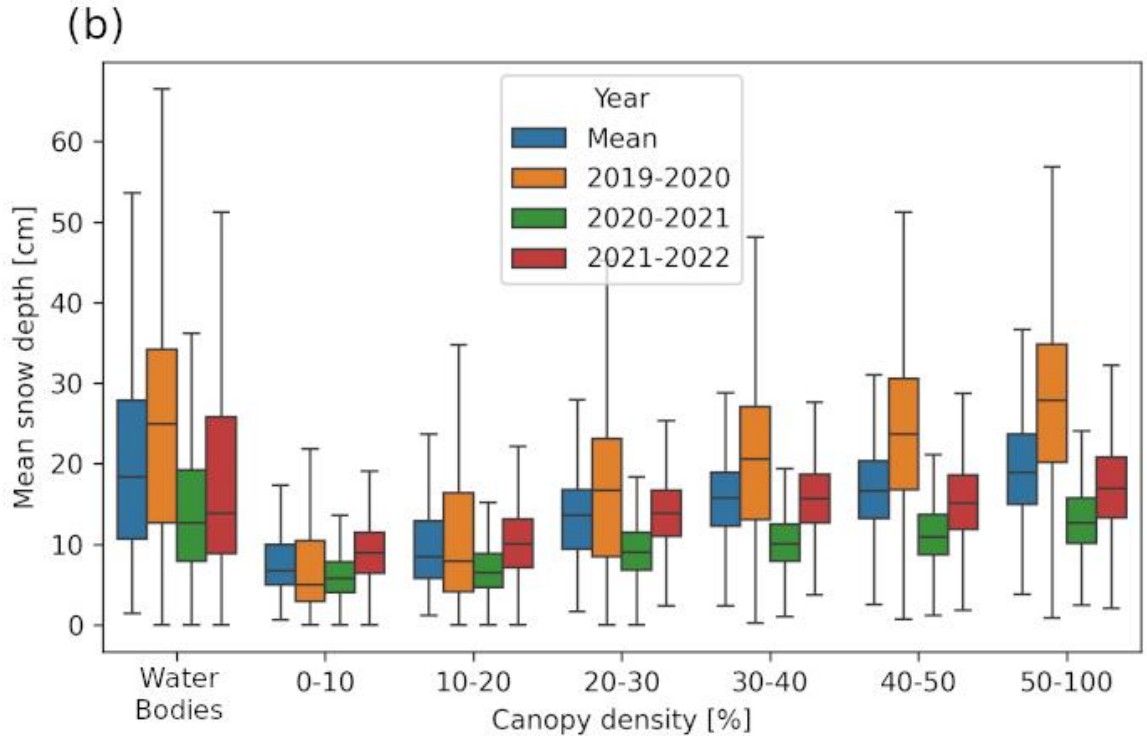

Figure 5: In situ measurements of snow depth compared to snow depth estimates derived from S1. Different colours represent
the different automatic weather stations, and the solid lines represent linear regressions of the dataset.

 Figure 6: In situ measurements of snow depth compared to snow depth estimates derived from S1. Different colours represent
 different years, and solid lines represent linear regression for each year.


Table 1: Mean snow depth values by the automatic weather stations (AWS), snow course measurements, and derived from the
S1 images separated by years.

| | AWS mean (cm) | | | | Sentinel-1 mean (cm) | | | |
|---|---|---|---|---|---|---|---|---|
| | **2019-2020** | **2020-2021** | **2021-2022** | **2019-2022** | **2019-2020** | **2020-2021** | **2021-2022** | **2019-2022** |
| IN | 53.7 ± 1 | 22.1 ± 1 | 35.5 ± 1 | **37.1 ± 1** | 31 ± 16 | 13.7 ± 8 | 14.8 ± 8 | **20 ± 11** |
| IK | 70.9 ± 1 | 28.3 ± 1 | 41.6 ± 1 | **46.9 ± 1** | 8.5 ± 7 | 11.6 ± 6 | 10.2 ± 7 | **10.1 ± 7** |
| IA | 61.7 ± 1 | 28.1 ± 1 | 44.9 ± 1 | **44.9 ± 1** | 16.3 ± 12 | 8.8 ± 6 | 15.4 ± 9 | **13.4 ± 9** |
| Overall | **56.6 ± 1** | **22.4 ± 1** | **38 ± 1** | **39 ± 1** | **18.6 ± 12** | **11.3 ± 7** | **13.5 ± 8** | **14.5 ± 9** |
| | Snow Courses (cm) | | | | | | | |
| IN | 57.3 ± 6 | 28.1 ± 3 | 45.0 ± 5 | **43.4 ± 5** | 45.8 ± 34 | 8.5 ± 9 | 5.7 ± 6 | **20.0 ± 16** |
| IR | 87.2 ± 10 | 52.5 ± 6 | 69.2 ± 8 | **69.6 ± 8** | 48.1 ± 25 | 16.8 ± 17 | 10.4 ± 10 | **25.1 ± 17** |
| IL | 91.8 ± 10 | 59.8 ± 7 | 68.7 ± 8 | **73.4 ± 8** | 24.6 ± 20 | 11.1 ± 11 | 16.0 ± 16 | **17.2 ± 16** |
| IA | 74.4 ± 8 | 39.6 ± 4 | 51.3 ± 6 | **55.1 ± 6** | 34.3 ± 56 | 23.9 ± 24 | 16.2 ± 16 | **24.8 ± 32** |
| IM | 67.1 ± 7 | 41.1 ± 5 | 38.0 ± 4 | **48.7 ± 5** | 47.6 ± 22 | 15.5 ± 15 | 11.3 ± 11 | **24.8 ± 16** |
| IKa | 93.3 ± 10 | 38.7 ± 4 | 49.4 ± 5 | **60.5 ± 7** | 9.8 ± 11 | 21.6 ± 22 | 18.6 ± 19 | **16.6 ± 17** |
| Overall | **78.5 ± 9** | **43.3 ± 5** | **53.6 ± 6** | **58.5 ± 6** | **35.0 ± 28.0** | **16.2 ± 16.2** | **13.0 ± 13.0** | **21.4 ± 19.1** |


Table 2: Mean absolute error (MAE) and root mean square error (RMSE) separated by years.

|  | MAE (cm) | RMSE (cm) |
|---|---|---|
| 2019-2020 | 38.9 | 48.6 |
| 2020-2021 | 14.0 | 18.7 |
| 2021-2022 | 25.5 | 32.7 |
| 2019-2022 | 26.1 | 35.6 |
