# Peer review of "Snow depth derived from Sentinel-1 compared to in-situ"

_EGUsphere, 2024_

## Author Response (AR1)

09$^{th}$ May 2025
Finnish Meteorological Institute
00560, Helsinki - FI

Dear Dr. Mottram,

**"Snow depth derived from Sentinel-1 compared to in-situ observations in northern Finland"**

Thank you for considering the above manuscript in The Cryosphere Discussions. Following the reviewers' feedback, we now resubmit a revised manuscript that addresses the suggestions raised.

In summary, we have (1) added further discussion relating to the interpretation of the data, (2) revised and expanded the main body of the manuscript, (3) included in situ snow courses measurements on 6 locations, (4) updated relevant figures, table, and added new ones.

We are grateful for the comments provided by the reviewers, as they have helped to improve the manuscript, and we hope that the changes are to their satisfaction.

Yours faithfully,

Adriano Lemos

(corresponding author)

**The responses (A) to the reviewer 1 (R1) and reviewer 2 (R2) are shown in blue below.**

R1: This paper used co- and cross-polarized backscatter data from Sentinel-1 SAR C-band images to estimate snow depth variations over northern Finland from 2019 to 2022. From the report of this paper, snow depth estimated from Sentinel-1 images tended to underestimate compared to measurements from automatic weather stations. Additionally, snow depth increased with higher canopy density. Their findings provide technical and theoretical references for estimating snow depth from C-band SAR images to some extent. However, several issues still need to be resolved before publication.

A: We thank the reviewer 1 for the time spent and for all the suggestions made, which have significantly improved the manuscript. Please find all the responses to the suggestions below.

**Comments and suggestions:**

R1: Line 57, it is suggested to provide a brief introduction about the limitations in estimating snow depth using C-band SAR.

A: We included additional text on the limitations of using C-band SAR data to estimate snow depth, emphasizing the complexity of the behaviour of the SAR backscatter inside the snowpack.

R1: Line 58, the abbreviation S1 has been provided here; henceforth, please use the abbreviation when referring to Sentinel-1.

A: We made the changes.

R1: Line 91-92, please provide descriptions of the terrain characteristics, meteorological conditions, vegetation cover, and other relevant information for these three weather stations.

A: We included additional information available for the AWSs. The elevation where the stations are located and forest cover index, extracted from the Multi-source National Forest Inventory Raster Maps of 2021.

R1: Line 124, for the empirical methodology developed by Lievens et al. (2019), in comparison to the original version, what specific improvements did this study make?

A: Similar comment from R2. The main contribution of the study is to expand the usage of an adapted version of the empirical methodology to estimate snow depth using S1 developed by Lievens et al (2019), and assess it using independent in situ measurements. We included a note in the methodology section that the adaptation is in the ratio "$\sigma^0_{vh}/\sigma^0_{vv}$". The ratio in our manuscript is done in dB, instead of a ratio in linear

scale. To support our choice, we generated the figures below showing different testing scenarios; Fig-i using the adapted version with different "a" and "b" empirical indices; and Fig-ii the updated version of the retrieval algorithm (Lievens et al., 2022) with different "a", "b", and "c" empirical indices.

We decided to not use the update version due to the amount of spikes and high variability of the result.

It is important to emphasize that the primary objective of this manuscript is not to enhance or refine the existing methodology. Rather, the intention is to evaluate and extend the application of an established methodology, incorporating certain adaptations.

i)                                                    ii)

[Figure]

R1: Line 166, this paper states "Overall, we observed clear underestimations in the shallow snow depth regions". However, Figure 2 clearly demonstrates a more pronounced underestimation in regions characterised by deep snow depth. Please provide possible reasons.

A: In fact, we observe underestimation for shallow and deep snow depth. We corrected the sentence in the text.

R1: Please delineate the boundaries of the water bodies in Figure 1.

A: The lakes are now delineated.

R1:There may have been rapid changes in snow depth during the period from April 3 to 7, 2002. It is not appropriate to compare the measured snow depth data from this period with the estimated snow depth from S1 on April 6.

A: We agree it is not appropriate to compare the snow pits measurements with the S1 estimates due to the rapid changes during the period. This is the main reason why the plots were not added in the main manuscript. However, we decided to include them, as

well as the snow courses measurements, in the supplement material to illustrate other in situ measurements sources available around the region.

R1: L193, it is confused to state "thicker snow depth values over dense vegetation and water bodies areas, where the canopy density is equal to 0%". The canopy density is equal to 0% means there is no vegetation.

A: We changed the Figure 4 x-axis label. We replaced "0%" to "Water Bodies" to avoid confusion.

R1: Line 197, change "Figure 5b" to "Figure 4b".

A: Corrected.

R1: L213-214, it is suggested to analyse the reasons why the correlation of the year 2019-2020 and 2021-2022 is lower than that of 2020-2021?

A: As we have no vertical profile measurements available of the snowpacks at the AWS sites, or parameters describing the state of the underlying soil, the response here is admittedly postulative. However, we know that for shallow seasonal snowpacks, the C-band SAR signal's penetration depth is sufficient to allow for "interference" from the underlying soil - or the snow/ice interface in cases of snow on lake or sea ice (Feng et al., 2021). Given that the methodology we applied was applied equally for all studied years, the most likely reason for the temporal SD correlation variability would seem to be variable inclusions of soil "signal" in the SAR data. Whether or not the underlying primary cause is the freeze status of the top soil layers is a question that we currently do not have data to answer. However, in future studies it would seem advisable to conduct validation experiments for shallow snow retrievals that also account for the properties of the interface beneath the snowpack. This line of thought is further reinforced by our finding that the retrieval underestimates snow depth particularly strongly over lake ice snow, where the snow-ice interface is even more complex from the EM signal scattering viewpoint than that of snow-soil.

- Feng, T., Hao, X., Wang, J., Li, H., & Zhang, J. (2021). Quantitative Evaluation of the Soil Signal Effect on the Correlation between Sentinel-1 Cross Ratio and Snow Depth. Remote Sensing, 13(22), 4691. https://doi.org/10.3390/rs13224691

R1: Line219-220, to achieve more accurate estimation of snow depth, what improvements should be considered for the estimation method?

A: We recognize that deriving snow depths using C-band SAR images over this region is still a challenge and further investigation is necessary. Moreover, given the under- and overestimations observed against reference snow depth data, we suggest that rigorous

radiative transfer model-based studies would be necessary to better understand the drivers of SAR backscatter from snowpacks.

The suggestions above are in the Conclusions section.

**Reviewer 2:**

This manuscript estimated snow depth over the northern Finland area from Sentinel-1 observation utilizing an existing retrieval algorithm, and compared the estimation with in-situ measurements from ground stations. Overall, the subject this study deals with is relevant to this journal, and the manuscript was easy to read and concise. However, the main issue was difficulty in capturing the main message or contributions that the authors are trying to deliver with this study.

A: We thank the reviewer 2 for the time spent and for all the suggestions made, which have significantly improved the manuscript. Please find all the responses to the comments below.

**General comments:**

R2: I thought the main contribution might be an algorithm development. However, the authors utilized an adapted version of Lievens et al. (2019), but it was not clearly stated which components of the original algorithm were adapted (modified/changed). How did the authors improve the algorithm? How did the modification affect the retrieval performance in reference to the original algorithm?

A: Similar comment from R1. The main contribution of the study is to expand the usage of an adapted version of the empirical methodology to estimate snow depth using S1 developed by Lievens et al (2019), and assess it using independent in situ measurements. We included a note in the methodology section that the adaptation is in the ratio "$\sigma^0_{vh}/\sigma^0_{vv}$". The ratio in our manuscript is done in dB, instead of a ratio in linear scale. To support our choice, we generated the figures below showing different testing scenarios; Fig-i using the adapted version with different "a" and "b" empirical indices; and Fig-ii the updated version of the retrieval algorithm (Lievens et al., 2022) with different "a", "b", and "c" empirical indices.

We decided to not use the update version due to the amount of spikes and high variability of the result.

It is important to emphasize that the primary objective of this manuscript is not to enhance or refine the existing methodology. Rather, the intention is to evaluate and extend the application of an established methodology, incorporating certain adaptations.

i)                                          ii)

[Figure]

R2: On the other hand, the comparison method was not rigorous enough to make this work a validation study. Weather station observation would represent the snow depth a few meters around the station, while the Sentinel-1-derived snow depth has much coarser resolution (it even went through a 990 m by 990 m kernel moving mean filter). How could those data be comparable? What was the authors' strategy to adequately address this issue?

A: You are correct. It's challenging to do a straight comparison between the coarse resolution data product from S1 (~1km), and the localized and punctual measurements of the AWSs. Unfortunately, despite all the efforts, the AWSs measurements are sparse and limited in space.

The upscaling of the S1 data has presented better snow depth estimates in previous studies (Hoppinen et al., 2024; Lievens et al., 2022), due to the fact that the data product becomes noisier and more variable without the speckle noise filtering (Dunmire et al., 2024). We included additional text emphasising this matter in the methodology part.

However, to enrich the comparison analysis, we included 6 locations of snow course (SC) measurements. Thanks to SYKE and FMI, there are also available around Finland the SC measurements. We added snow depth measurements (Figure S3) from SC at 6 different locations (Figure 1) available for the region (description included in the section 2). These are averaged snow depth measurements every 50 meters, along 2-4km for each SC.

R2: In addition, while there are many hypotheses for snow depth underestimation, the supporting evidence was rarely provided. The authors mention that temperature and precipitation are important factors, but they were not taken into account in the analysis. The retrieval algorithm also doesn't consider temperature. Could this be one of the reasons for underestimation?

A: There is perhaps a misunderstanding here; while temperature and precipitation are strong drivers of snow depth in general, these parameters are not incorporated in the SAR-based snow depth estimation. In principle one could design a satellite-based retrieval which takes advantage of a priori available T and precipitation, but due to scarcity of robust, validated data on precipitation we have not attempted such an approach here.

R2: Lastly, the review of previous studies seems to be incomplete. For example, I have the following questions. Is this the first Sentinel-1 snow depth retrieval study that covers northern Finland? Is Lievens et al. (2019) the only C-band SAR snow depth retrieval algorithm? Why was this algorithm chosen for this study? How good or bad are Sentinel-1 snow depth retrievals over different areas or regions based on different algorithms? Are there any previous studies/reports available?

A: We included additional information and other review studies in the manuscript (list below). However, a much better recollection of SAR methods and techniques can be found in some review papers, included in the manuscript (Tsai et al., 2019; Awasthi & Varade, 2020; Tsang et al., 2022).

- Awasthi, S., & Varade, D. (2021). Recent advances in the remote sensing of alpine snow: a review. In GIScience and Remote Sensing (Vol. 58, Issue 6, pp. 852–888). Taylor and Francis Ltd. https://doi.org/10.1080/15481603.2021.1946938.
- Dunmire, D., Lievens, H., Boeykens, L., & de Lannoy, G. J. M. (2024). A machine learning approach for estimating snow depth across the European Alps from Sentinel-1 imagery. Remote Sensing of Environment, 314. https://doi.org/10.1016/j.rse.2024.114369.
- Hoppinen, Z., Palomaki, R. T., Brencher, G., Dunmire, D., Gagliano, E., Marziliano, A., Tarricone, J., & Marshall, H.-P. (2024). Evaluating snow depth retrievals from Sentinel-1 volume scattering over NASA SnowEx sites. The Cryosphere, 18(11), 5407–5430. https://doi.org/10.5194/tc-18-5407-2024.
- Sun, S., Che, T., Wang, J., Li, H., Hao, X., Wang, Z., & Wang, J. (2015). Estimation and analysis of snow water equivalents based on C-band SAR data and field measurements. Arctic, Antarctic, and Alpine Research, 47(2), 313–326. https://doi.org/10.1657/AAAR00C-13-135.
- Tsai, Y. L. S., Dietz, A., Oppelt, N., & Kuenzer, C. (2019). Remote sensing of snow cover using spaceborne SAR: A review. In Remote Sensing (Vol. 11, Issue 12). MDPI AG. https://doi.org/10.3390/rs11121456.

We are not sure if this manuscript is the first S1 snow depth retrieval in Northern Finland. At least, we could not find any other study of snow depth estimates in Finland using Sentinel-1. If there are others, we are not aware of them, and we are happy to accept the indication.

Lievens group published the methodology in 2019, and an amended version in 2022. At the time of submission (late 2023), those were the only ones we were aware of. Currently, there are some other studies, which were added to the main manuscript.

**Specific comments:**

R2: L194: There are two periods ("..") next to "40".

A: corrected.

R2: Figure 1: Please consider changing the color of letters (currently in black) over the dark blue background. It is difficult to read.

A: In Figure 1 we changed the colour of letters, the colour of the symbols, and delineated the lake regions.